# Transcriptional Readthrough Interrupts Boundary Function in Drosophila

**DOI:** 10.3390/ijms241411368

**Published:** 2023-07-12

**Authors:** Olga Kyrchanova, Vladimir Sokolov, Maxim Tikhonov, Galya Manukyan, Paul Schedl, Pavel Georgiev

**Affiliations:** 1Department of the Control of Genetic Processes, Institute of Gene Biology Russian Academy of Sciences, 34/5 Vavilov St., Moscow 119334, Russia; olgina73@gmail.com (O.K.); volodjn@yandex.ru (V.S.); 2Center for Precision Genome Editing and Genetic Technologies for Biomedicine, Institute of Gene Biology, Russian Academy of Sciences, 34/5 Vavilov St., Moscow 119334, Russia; me@mtih.me (M.T.); galya_manukyan@mail.ru (G.M.); 3Department of Molecular Biology, Princeton University, Princeton, NJ 08544, USA

**Keywords:** chromatin boundary, insulator, bithorax complex, Abd-B, scs, Fub, Fab-7, transcription, lncRNA

## Abstract

In higher eukaryotes, distance enhancer-promoter interactions are organized by topologically associated domains, tethering elements, and chromatin insulators/boundaries. While insulators/boundaries play a central role in chromosome organization, the mechanisms regulating their functions are largely unknown. In the studies reported here, we have taken advantage of the well-characterized *Drosophila* bithorax complex (BX-C) to study one potential mechanism for controlling boundary function. The regulatory domains of BX-C are flanked by boundaries, which block crosstalk with their neighboring domains and also support long-distance interactions between the regulatory domains and their target gene. As many lncRNAs have been found in BX-C, we asked whether readthrough transcription (RT) can impact boundary function. For this purpose, we took advantage of two BX-C boundary replacement platforms, *Fab-7^attP50^* and *F2^attP^*, in which the *Fab-7* and *Fub* boundaries, respectively, are deleted and replaced with an *attP* site. We introduced boundary elements, promoters, and polyadenylation signals arranged in different combinations and then assayed for boundary function. Our results show that RT can interfere with boundary activity. Since lncRNAs represent a significant fraction of Pol II transcripts in multicellular eukaryotes, it is therefore possible that RT may be a widely used mechanism to alter boundary function and regulation of gene expression.

## 1. Introduction

The chromosomes of multicellular animals are organized into a series of looped domains called TADs (topologically associated domains) [1,2,3,4]. While a variety of elements contribute to folding the chromatin fiber (e.g., the tethering elements that help link enhancers to promoters [5]), this three-dimensional organization depends, in part, on special elements called boundaries or insulators [1,6,7]. Although boundary elements have now been identified in many different species, they have been most thoroughly characterized in *Drosophila* [1,7,8].

Fly boundaries are 150–1500 base pairs (bp) in length and span one or more nucleosome-free nuclease hypersensitive regions that are formed by different combinations of chromosomal architectural proteins, including *Drosophila* CTCF (dCTCF) [9,10,11].

Functional studies using transgene assays indicate that fly boundary elements function as insulators [7,12,13]. When placed between an enhancer or silencer and a reporter, they prevent regulatory interactions. When a reporter is bracketed by boundary elements, they protect against chromosomal position effects. With some exceptions, boundary function in these assays is “constitutive”—i.e., it is observed throughout development and is independent of cell type. The likely reasons for this constitutive activity are that most of the fly architectural proteins are ubiquitously expressed [8] and that different combinations of these proteins are deployed to generate the activity of individual boundaries [10,14].

Since multiple functionally redundant architectural proteins contribute to the functions of individual fly boundaries in flies, it seems unlikely that boundaries will undergo genome-wide reorganization during cellular differentiation as this would require a change in the patterns of expression of multiple chromosomal proteins. Rather, one might expect that boundary organization would be subject to local alterations by modulating the insulating functions of specific boundary elements. In the studies reported here, we have used the *Drosophila* bithorax complex (BX-C) to identify potential mechanisms for modulating local boundary function.

BX-C is responsible for specifying the nine posterior-most parasegments (PS5–PS14 in embryos, segments T3–A9 in adults) of flies [15,16,17,18]. Since there are three homeotic genes in BX-C: *Ultrabithorax* (*Ubx*), *abdominal-A* (*abd-A*), and *Abdominal-B* (*Abd-B*) (Figure 1A), the regulation of their expression in each parasegment determines the correct segment differentiation. This is accomplished by subdividing the complex into nine *cis*-regulatory domains. Each domain has tissue- and stage-specific enhancers responsible for directing a unique parasegment-specific pattern of expression of one of the homeotic genes [16,19,20,21,22,23]. *Ubx* is responsible for specifying PS5 (T3) and PS6 (A1), and its expression in these two parasegments is controlled by the *bx*/*abx* and *bxd*/*pbx* regulatory domains, respectively. The *infra-abdominal* (*iab*) domains regulate the transcription of *abd-A* and *Abd-B*. The *abd-A* gene is controlled by *iab-2*, *iab-3*, and *iab-4* in PS7 (A2), PS8 (A3), and PS9 (A4), respectively. Four domains, *iab-5*, *iab-6*, *iab-7*, and *iab-8,9*, regulate *Abd-B* expression in PS10 (A5), PS11 (A6), PS12 (A7), and PS13,14 (A8 (♀), A9 (♂)), respectively (Figure 1A).

The regulatory domains are activated sequentially in successive parasegments along the anterior–posterior axis [17]. The activity state, *on* or *off*, of each regulatory domain is set early in development by maternal, gap, and pair-rule gene proteins, which bind to initiation elements in each domain [24,25,26]. For example, (Figure 1B), in PS11 (A5), the *iab-5* initiator turns *on* the *iab-5* domain, while the adjacent *iab-6* and other more distal (relative to the centromere) domains remain in the *off* state. In PS11 (A6), the initiator in *iab-6* turns the *iab-6* domain *on*. While *iab-5* is also active in PS11, *iab-7* is *off*. Once the activity state is set, it is remembered during the remainder of development by mechanisms that depend upon trithorax and Polycomb group proteins [27,28,29,30]. Each domain is flanked by boundary elements which function to block crosstalk between initiation elements in adjacent regulatory domains [26,27,31,32,33,34,35,36]. For example, the *Fab-7* boundary in the *Abd-B* region of BX-C separates the *iab-6* and *iab-7* domains. When *Fab-7* is deleted, *iab-6* and *iab-7* fuse into a single domain, and the *iab-6* initiation element inappropriately activates *iab-7* in PS11 (A6) (Figure 1C). As a result, *iab-7* drives *Abd-B* expression not only in PS12 (A7) but also in PS11 (A6), inducing a gain-of-function (GOF) phenotype, manifested in the transformation of transforming PS11 (A6) into a copy of PS12 (A7).

Some BX-C boundaries also have a second function, which is boundary “bypass” [37]. For example, the *Abd-B* regulatory domains *iab-5*, *iab-6*, and *iab-7* are separated from their gene target by one or more boundaries (Figure 1A). In order for these domains to regulate *Abd-B,* there must be a mechanism that enables the enhancers in each domain to bypass the intervening boundaries. Recent studies have shown that the *Fab-7* and *Fab-8* boundaries have subelements that confer bypass activity and enable the domain immediately distal to the boundary to “jump over” the intervening boundaries and activate *Abd-B* expression [38,39,40]. This function is important in understanding the phenotypes that are observed when *Fab-7* is replaced by a heterologous boundary lacking bypass activity. Typically, the heterologous boundary can block the *iab-6* initiators from activating *iab-7* in PS11/A6. This rescues the GOF transformation that is observed when the *Fab-7* boundary is deleted. However, because the heterologous boundary lacks bypass activity, it interferes with interactions between the *iab-6* regulatory domain and *Abd-B* in PS11/A6. As a result, *iab-6* is not able to drive the appropriate level of *Abd-B* expression in PS11/A6, and it takes on morphological features characteristic of the more anterior parasegment (segment) PS10 (A5).

In previous studies [41,42], the *Fab-7* boundary was replaced with two versions of the *scs* insulator from the 87A7 heat shock locus, *scs,* and *scs^min^* [43,44,45]. The larger version, *scs*, is a complex insulator containing the *Cad87A* and *CG31211* promoters (Figure 2A). The smaller fragment, *scs^min^*, lacks most of the *Cad87A* promoter [45] (Figure 2A). Hogga et al., 2002, showed that when the *Cad87A* promoter in the larger *scs* replacement is oriented towards *iab-6,* it disrupts the functioning of the *iab-6* and *iab-5* regulatory domains inducing a loss-of-function (LOF) phenotype in which the A6 and A5 cuticle had morphological features like A4 [41]. They suggested that RT from the *Cad87* promoter inactivates enhancers in *iab-5* and *iab-6* required for the development of the adult cuticle. However, a completely different result was observed when the *Cad87* promoter in the *scs* replacement was oriented towards *iab-7*. In this case, a GOF phenotype was induced: A6 was converted into a copy of A7. To explain this result, it was suggested that transcription into *iab-7* disrupted Polycomb-dependent silencing but did not impact the activity of the *iab-7* tissue/stage-specific enhancers [45]. As the explanations for the phenotypic effects of the *scs* replacement in the two different orientations seemed to be inconsistent, we decided to reinvestigate the functioning of both *scs* and *scs^min^* replacements in *Fab-7*. In the course of our studies, we discovered that RT appears to be a general mechanism for turning boundary and enhancer functions off and changing patterns of gene regulation.

## 2. Results

### 2.1. The scs^min^ Insulator Can Block Crosstalk between the iab-5 and iab-6 Domains Only in Cooperation with the iab-7 PRE

In the boundary replacement experiments of Hogga et al. [41,42], the *scs^min^* insulator was introduced into a *Fab-7* deletion which removes three of the four nuclease hypersensitive sites associated with the *Fab-7* boundary, HS*, HS1, and H2. This deletion results in an incomplete GOF transformation of A6 (PS11) into A7 (PS12) [36]. When *scs^min^* replaced this deletion, it blocked crosstalk between the *iab-6* and *iab-7* initiation elements and rescued the GOF phenotype of the starting deletion. However, since *scs^min^* does not support boundary bypass, the A6 segment was transformed towards A5 [42].

In the *Fab-7* deletion used by Hogga et al. [41,42], the HS3 sequence was still present because it was thought at that time that HS3 corresponded to the *iab-7* Polycomb response element (*iab-7* PRE) [36,47,48] and that its only function was Polycomb-dependent silencing. However, recent work has shown that HS3 has insulating activity [46] and that this, not silencing, is its primary function. In fact, a fully functional *Fab-7* boundary can be generated by combining HS3 with the distal half of HS1 (dHS1) [38,46]. This finding made us wonder whether *scs^min^* would be able to block crosstalk between *iab-6* and *iab-7* in the absence of HS3.

To address this question, we used a previously characterized *Fab-7^attP50^* replacement platform in which the HS*, HS1, HS2, and HS3 are substituted by an *attP* site [49] (Figure 2B). In the starting *Fab-7^attP50^* deletion, *iab-7* is inappropriately activated in A6 (PS11), and this results in the transformation of A6 into a copy of A7 so that both the A6 and A7 segments are absent in adult males (Figure 2C).

To test for *scs^min^* function with and without HS3, we introduced two replacements, *scs^min^+HS3* and *scs^min^*, into the *Fab-7^attP50^* platform. Though *scs^min^+HS3* has a slightly different sequence composition compared to the *scs^min^* replacement published by Hogga et al. [42] due to the use of different replacement platforms, their activity is similar. As shown in Figure 2C, the *scs^min^+HS3* combination rescues the GOF transformation of A6 in adult males, and the A6 segment is present. However, because this combination lacks bypass activity, the *iab-6* regulatory domain is blocked from regulating *Abd-B* expression in A6. As a consequence, the morphology of the A6 segment resembles that of A5. Instead of a banana shape without any bristles, the sternite has a quadrilateral shape and is covered in bristles, and it resembles the sternite in A5. In *wild type* (*wt*), the trichome hairs on the A6 tergite are restricted to the anterior and ventral margin, while the trichome hairs cover almost all of the A5 tergite (Figure 2C). As can be seen in the darkfield image, the A6 tergite in *scs^min^+HS3* males is covered with trichome hairs just like A5. In addition, there are patches of unpigmented tergite in A6 and also A5; these transformations will be considered further below. As was observed for *Fab-7* (class II) deletions that retained HS3 [36], the *HS3* replacement alone has only a limited ability to block crosstalk between *iab-6* and *iab-7* [46]. In *HS3* males, the A6 tergite is greatly reduced in size, and the A6 sternite is completely missing (Figure 2C).

Like *HS3*, the *scs^min^* replacement only partially blocks crosstalk between *iab-6* and *iab-7* (Figure 2C). However, it differs from the *HS3* replacement in that there is a range of phenotypes in adult *scs^min^* males. In all *scs^min^* males, there is a residual A6 tergite, while the A6 sternite is absent. The residual tergite has patches of cells with trichome hairs indicating that in these cells, there is a LOF transformation in parasegment/segment identity from PS11/A6 to PS10/A5. In about 30% of the males, the morphogenesis of A5 is also affected. As shown in Figure 2C, there are patches of tissue in the A5 tergite that are not fully pigmented. This phenotype indicates that the *iab-5* regulatory domain is not fully functional in a subset of *scs^min^* males, and we will return to this issue below.

### 2.2. Transcription Induced by the Cad87A Promoter in scs Can Affect the Activity of the iab-7 Domain

The finding that *scs^min^* must be combined with HS3 to block crosstalk between *iab-6* and *iab-7* efficiently prompted us to examine the blocking activity of the larger *scs* fragment, which contains the *Cad87A* promoter (Figure 3A).

Initially, we inserted *scs* in the reverse orientation in the *Fab-7^attP50^* platform (*scs^R^*). As shown in Figure 3B, *scs^R^* males have an unusual LOF phenotype in which A7 is transformed towards A6. Unlike *wt* males, which lack an A7 segment, *scs^R^* males have an A7 tergite and sternite. The tergite is fully pigmented, and the trichome hairs are largely (but not completely) restricted to the anterior and lateral edges. The sternite has a banana shape like the sternite in A6; however, it is malformed and has bristles. One explanation for this unexpected phenotype is that transcription from the *Cad87A* promoter into the *iab-7* domain may lead to an inactivation of the *iab-7* enhancers. To test this possibility, we inserted a transcription termination/polyadenylation signal (*PAS*) in between *scs^R^* and the *iab-7* domain. Figure 3B shows that *scs^R^+PAS* males lack an A7 segment. This finding indicates that when readthrough transcription from *Cad87A* is blocked, the *iab-7* enhancers are functional (Figure 3B). On the other hand, the *scs* insulator is at most partially functional: there is a rudimentary A6 tergite with patches of ectopic trichome hairs, while the A6 sternite is absent.

### 2.3. Transcription Disrupts Boundary Function

The findings in the previous section indicate that, like *scs^min^*, *scs^R^* is unable to effectively block crosstalk between *iab-6* and *iab-7* on its own. We wondered whether the larger *scs^R^* element would be functional when combined with HS3 or whether *Cad87A*-dependent transcription would disrupt the insulator function of *HS3* just like it does for the *iab-7* enhancers. For this purpose, we generated a tripartite combination of *scs^R^+HS3+PAS* (Figure 3A). The *PAS* sequence was included to protect the *iab-7* enhancers from transcriptional readthrough. Figure 3B shows that male flies carrying the *scs^R^+HS3+PAS* combination have a GOF transformation of A6 towards A7, just like that observed for *scs^min^*, indicating inactivation of the insulator function. This result suggests that transcription initiated at the *Cad87* promoter adversely impacts the insulator activity of HS3 (Figure 3B).

To further test the idea that RT can disrupt boundary function, we generated a quadripartite replacement, *5′P+scs^min^+HS3+PAS*, including the P-element promoter (*5′P*). As shown in Figure 3B, the inclusion of the P-element promoter disrupts the boundary activity of *scs^min^+HS3*. While *scs^min^+HS3* on its own rescues the boundary between the *iab-6* and *iab-7* (Figure 2C), this is not true for *5′P+scs^min^+HS3+PAS* (Figure 3B). When the P-element promoter is included in the replacement, the A6 segment is almost completely absent, as would be expected if RT disrupts insulator function.

### 2.4. Transcription Disrupts the Functioning of the iab-6 and iab-5 Regulatory Domains

The phenotypic abnormalities of *scs^R^* are not restricted to A7 (Figure 3B). The A6 tergite has ectopic patches of trichome hairs, while the A6 sternite has bristles and is misshapen. Though A6 shows evidence of a LOF transformation in portions of the adult cuticle, the opposite effect is observed in A5. The A5 tergite of *scs^R^* is partially devoid of trichome hairs, while A5 sternite has an abnormal banana-like shape. In *scs^R^+PAS* and *scs^R^+HS3+PAS* lines (Figure 3B), the male A5 segment is also affected: the sternite is reduced in size, while tergite has variable pigmentation and patches of ectopic trichome hairs (mix LOF/GOF phenotype). The effects of *scs^R^* on A5 (and also A6) suggest that transcription from the *CG31211* promoter is disrupting the functioning of *iab-5* (and *iab-6*). To test this idea, we generated a *PAS+scs^R^+HS3+PAS* transgenic line in which a second *PAS* sequence was inserted between the *iab-6* domain and *scs^R^* (Figure 3B). In *PAS+scs^R^+HS3+PAS* males, the A5 segment displays a nearly *wt* phenotype. This finding suggests that transcription initiated from the *CG31211* promoter perturbs the proper development of the A5 segment.

It seemed possible that a similar mechanism, namely transcription from the residual part of the *Cad87A* promoter, might account for the variable and unexpected LOF defects in the development of A5 that we observed (see above) in the *scs^min^* replacement (Figure 2C). To further explore the effects of transcription on the functioning of the *iab-5* and *iab-6* regulatory domains, we inserted *scs* in the forward orientation. We generated three different insertions, *scs* alone, *scs* plus HS3 (*scs+HS3*), and *scs* plus the three major *Fab-7* hypersensitive sites, HS1, HS2, and HS3 (*F7^HS1+2+3^*) (Figure 4A). The *scs+HS3* and *scs+F7^HS1+2+3^* combinations have blocking activity and rescue the GOF transformations evident in the starting *F7^attP50^* platform. On the other hand, consistent with the idea that transcription can disrupt the activity of the *iab-5* and *iab-6* enhancers, we find that in both replacements, A6 and A5 have an A4-like phenotype. This is most clearly seen in the pattern of pigmentation and in the dense trichome hairs in the A5 and A6 tergites (compare A5 and A6 with A4 in Figure 4B).

A more complicated phenotype is observed with *scs* alone (Figure 4B). As expected, blocking activity is not complete, and A6 shows evidence of GOF transformations. The A6 tergite is reduced in size, while there is only a small patch of sternite tissue. In both cases, the residual A6 tissue has a phenotype indicative of a transformation towards A4 identity: there are bristles on the patch of sternite tissue, while the residual tergite is depigmented and has patches of ectopic trichome hairs. Interestingly a mixed GOF and LOF phenotype is also observed in A5: both the sternite and tergite are reduced in size as expected for a GOF transformation, while the tergite is depigmented, and there are patches of densely packed trichome hairs. The GOF transformations in the *scs* replacement resemble those seen when both *Fab-7* and *Fab-6* boundaries are deleted (see *F6^attP^+F7^attP^* in Figure 4B). There is also evidence of a weak GOF transformation of A4 in the double boundary deletion. However, unlike *scs,* the double boundary deletion shows no evidence of LOF transformations of A5 and A6.

These findings indicate that transcription from the *Cad87* promoter in *scs* directed towards *iab-6* and *iab-5* disrupts not only the boundaries but also the functioning of enhancers of these domains. They also suggest that the variable LOF phenotypes in A5 evident in males carrying the *scs^min^* replacement might be due to a low level of transcription from the truncated *Cad87A* promoter. Since the LOF phenotypes in A5 varied between individuals and were seen in only about 30% of the *scs^min^* males, one plausible explanation is that stochastic differences in promoter activity between individuals might account for incomplete penetrance. To test this possibility, we generated a *PAS+scs^min^* replacement (Figure 4C). Unlike *scs^min^*, the A5 tergite in *PAS+scs^min^* males is fully pigmented in all adult males, which would suggest that transcription from the clipped *Cad87* promoter is likely responsible for the pigmentation defects in A5. However, this does not seem to be true for the trichome hairs on the tergite, as they are still densely packed like those in A4. This finding indicates that the trichome hair phenotype is likely due to the blocking activity of the *scs^min^* element, which prevents *iab-5* from regulating *Abd-B* in cells that can give rise to trichome hairs.

To further investigate the effects of RT, we placed *HS3* upstream of *scs^min^* in *HS3+scs^min^*. Unlike *scs^min^+HS3*, *HS3+scs^min^* is unable to prevent crosstalk between *iab-6* and *iab-7*, and A6 is transformed towards A7 (compare *scs^min^+HS3* with *HS3+scs^min^* in Figure 4C). However, *HS3* is able to complement *scs^min^* when RT is blocked by an interposed *PAS* sequence (*HS3+PAS+scs^min^*, Figure 4C). Thus, a low level of transcription from the truncated *Cad87A* promoter is apparently sufficient to impact the boundary activity of *HS3*.

### 2.5. Readthrough Transcription Disrupts the Functioning of a Minimal Fub (pHS2) Replacement Boundary

We wondered whether RT would also impact the functioning of other boundary elements. To investigate this possibility, we chose the BX-C *Fub* boundary. *Fub* marks the border between the *Ubx* regulatory domain, *bxd*/*pbx*, and the *abd-A* gene and its regulatory domain, *iab-2* [32]. As illustrated in Figure 5A, there are two *Fub* hypersensitive regions, HS1 and HS2. The larger *Fub* hypersensitive region HS2 contains motifs for several known chromosomal architectural proteins. The distal 177 bp HS2 sequence (*dHS1*) has binding sites for dCTCF and Su(Hw), and we found that it can function as an effective boundary [14,50]. The proximal 450 bp HS2 sequence (*pHS2*, Figure 5A) contains binding sites for Pita and Su(Hw) [51,52].

We first tested whether *pHS2* is able to function as a boundary when introduced into the *Fab-7^attP50^* platform. As shown in Figure 5B, *Fub pHS2* has an insulator function and rescues the GOF phenotype of the *Fab-7^attP50^* deletion. Like most other heterologous replacements, *pHS2* blocks crosstalk but does not support bypass: an A6 segment is present in the *pHS2* replacement; however, its morphological features indicate that it has an A5 rather than an A6 identity: the sternite is misshapen and is covered in bristles. While the A5 tergite is fully pigmented, the trichome hairs are densely packed, much like the A4 tergite (consistent with the idea that trichome morphology in A5 is more sensitive to blocking activity by replacement boundaries than pigmentation).

A different result is obtained when the P-element promoter is placed upstream of *pHS2* in the *Fab-7* replacement (Figure 5B). As was observed for the P-element combination *5′P+scs^min^+HS3*, *pHS2* boundary activity is lost in *5′P+pHS2,* and the A6 segment is missing. To test whether this is due to RT from the P-element promoter, we generated two additional replacement combinations. In the first, the *PAS* element was placed downstream of the *pHS2* boundary to give *5′P+pHS2+PAS*, while in the second, the *PAS* element was placed between the P-element promoter and *pHS2* to give *5′P+PAS+pHS2.* As would be expected if RT disrupts boundary function in the *5′P+pHS2+PAS* replacement, there is only a residual A6 tergite, while the insulator function is rescued when the *PAS* element is placed between the P-element promoter and *pHS2* (*5′P+PAS+pHS2*) (Figure 5B).

### 2.6. Readthrough Transcription Disrupts Fub Function in Its Endogenous Context

Bender and Fizgerald (2002) [53] generated a series of imprecise hop outs of a P-element transgene inserted near the distal end of the *bxd*/*pbx* regulatory domain close to the sequences that were subsequently found to correspond to the *Fub* boundary [32,53]. These hop-out events induced an anterior-to-posterior transformation of A1 towards A2 identity. Molecular characterization of one of the hop outs that had a particularly strong GOF phenotype, *Uab^HH1^*, revealed that it was a truncated P-element transgene that retained only the P-element promoter and 65 bp of *lacZ* coding sequence. The P-element transgene was also inverted so that the promoter was pointing towards the *Fub* boundary and the *abd-A* gene. Several potential mechanisms were proposed to account for the transformation of A1 to A2 induced by P-element transcription [53]. One was that transcription disrupted the functioning of an as-yet unidentified boundary that blocked crosstalk between the *Ubx bxd*/*pbx* and *abd-A iab-2* regulatory domains. A second was that transcription interfered with the functioning of an element in *iab-2* that is required to keep the *iab-2* domain silenced in A1.

To test the boundary model, we took advantage of a *Fub* replacement platform *F2^attP^* (Figure 6A) that removes a 2106 bp sequence containing the two nuclease hypersensitive sites associated with the *Fub* boundary and replacing it with an *attP* site (*F2^attP^*) [14]. As shown in Figure 6B, the A1 tergite in *wt* is narrower than the A2 tergite, lacks bristles, and has less pigmentation, while the A1 sternite is absent. In *F2^attP^* males, the A1 segment is transformed into a copy of A2: the tergite is larger, and it has a pigmentation and bristle pattern like A2, and there is also a sternite that is covered in bristles. These phenotypic transformations in the adult cuticle resemble those previously reported [53] for the P-element hop-out mutants.

The GOF transformations evident in the starting *F2^attP^* deletion platform can be fully rescued by a 1587 bp fragment, *F2*, which includes both HS1 and HS2 (Figure 6A). Consistent with the previous results [53], we find that rescuing activity is disrupted when the P-element promoter, *5′P*, is placed upstream of the *F2* fragment (*5′P+F2*). In this replacement, the A1 segment resembles A2, just like the starting deletion platform (Figure 6B). Since introducing the PAS element downstream of *F2* in the *5′P+F2+PAS* combination does not rescue the GOF transformation, it would appear that boundary function rather than a downstream silencing element is the critical target for transcription inactivation.

## 3. Discussion

Blocking the activity of *scs* is context dependent: Our results indicate that the *scs* boundary has only a limited ability to block crosstalk between the *iab-6* and *iab-7* regulatory domains. This result is unexpected, as in transgene assays, *scs* was found to have one of the “stronger” insulator activities [43,44,45,54,55]. It seems likely that *scs* is a poor match with the neighboring *Fab-6* and *Fab-8* boundaries [56,57]. Both depend upon CTCF, while *scs* does not [58]. Also, when placed in the context of BX-C, *scs* seems to have a cell and/or an enhancer-specific blocking activity. For example, the phenotype of the A5 tergite in *PAS*+*scs^min^* males (Figure 4C) suggests that *scs* is unable to block the regulatory interactions between *iab-5* and *Abd-B* required for *wt* pigmentation, while its insulating activity is sufficient to block the interactions needed to inhibit the formation of trichomes.

Transcription disrupts enhancer activity: We found that when *scs^R^* was introduced into a larger *Fab-7* deletion that lacks HS3, transcription from the *Cad87A* promoter interferes with the functioning of the *iab-7* domain, inducing a LOF transformation. The suggestion that RT can disrupt the activity of tissue-specific enhancers in this region of BX-C is supported by the effects of inserting *scs* in the direct orientation. In this case, it induces a LOF transformation of both A6 (PS11) and A5 (PS10) towards A4 (PS9). The effects of transcription from the *Cad87A* promoter on these regulatory domains are most clear-cut when *scs* is combined with *HS3* or *F7^HS1+2+3^*. In both of these replacements, the combination of *scs* with *HS3* or *F7^HS1+2+3^* suppresses the GOF phenotype of the *Fab-7^attP50^* deletion platform, making the LOF transformations in A6 (PS11) and A5 (PS10) more obvious.

Transcription disrupts boundary function: The enhancers in the *Abd-B* regulatory domains are not the only elements whose function is disrupted by RT. We find that boundary activity can also be abrogated by RT. In the case of our *Fab-7* replacements, this is most directly demonstrated when boundaries are placed downstream of a P-element promoter. The *scs^min^+HS3* combination not only rescues the GOF transformation of A6 (PS11) in the *Fab-7^attP50^* deletion platform but also prevents *iab-6* and, to a lesser extent, the *iab-5* domain from regulating *Abd-B* expression. However, if the P-element promoter is placed upstream of *scs+HS3* as in the *5′P+scs+HS3+PAS* combination, blocking activity is largely lost, and A6 is transformed towards an A7 identity.

The effects of transcription on boundary activity are not limited to *scs* and *HS3* (*Fab-7*), as transcription also interferes with the functioning of the *Fub* boundary fragment *pHS2*. On its own, it rescues the GOF phenotype of *Fab-7^attP50^*; however, when placed downstream of the P-element promoter, blocking activity is lost. Transcription also inactivates the 1587 bp *Fub* boundary in its native context (Figure 6). Moreover, as was the case for a combination in which the truncated *Cad87A* promoter in *scs^min^* is pointing towards *HS3*, the disruption in the *pHS2* boundary function by the P-element promoter can be rescued by placing the *PAS* element in between the promoter and the boundary. These findings argued that boundary function is disrupted by RT rather than some other properties of the promoter.

While previous studies have shown that RT can suppress the activity of enhancers and promoters, how this happens is not fully understood [59,60,61]. One idea is that RNA Pol II transiently displaces DNA-binding proteins as it passes [62,63]. In the case of boundary elements, it seems possible that even a transient displacement of factors important for their activity could have a significant impact on boundary function. Fly boundaries link distant sequences together to form looped domains or TADs by boundary: boundary pairing interactions [5,10,12]. In this mechanism, TADs are formed when proteins associated with one boundary element physically interact in a stable fashion with proteins associated with a second boundary element. This means that a transient displacement of boundary-associated proteins from one of the elements would disrupt the TAD as it would uncouple the physical linkage between the distant sequences that define the endpoints of the loop. Consistent with this idea, a low level of transcription from the truncated *Cad87A* promoter can disrupt the boundary functions of *HS3*.

While our experimental paradigm is artificial, there are contexts in which RT provides a mechanism for coordinating higher-order chromosome organization with regulating gene activity. For example, the blocking activity of the *Fub-1* boundary is turned off by RT of a lncRNA from a promoter that is activated by the *Ubx* regulatory domain *bxd*/*pbx* [64] in PS6/A1 and more posterior parasegments. Inactivation of the *Fub-1* boundary enables enhancers in the *bxd*/*pbx* domain to regulate *Ubx* expression. MicroC experiments suggest that RT of the *Fub-1* boundary is likely accompanied by a switch from one TAD configuration to another configuration. Since transcription is not continuous but instead occurs in bursts that can differ both in their length and frequency depending on the specific enhancer–promoter combinations, an RT mechanism would result in only a transient remodeling of the TAD organization. Moreover, this remodeling would also be subject to regulation. In this respect, it is interesting to note that lncRNAs are thought to account for a vast majority of the transcripts in mammalian genomes [65,66,67,68]. It would be reasonable to suppose that some of these lncRNAs span boundaries and “tethering” elements that might also be sensitive to RT. RT of these lncRNAs would then alter the local chromatin organization and, in doing so, generate new combinations of regulatory elements and potential target genes.

## 4. Materials and Methods

### 4.1. Generation of the Replacement Lines

The strategy of the *Fab-7* replacement lines is described in detail in [49] and is based on the *Fab-7^attB50^* landing platform in which the 1950 bp *Fab-7* region was deleted, as shown in Figure 1A. *Fab-7^attB50^* landing platform contains an *attP* site for the integration of the tested constructs; *lox* and *frt* sites were used for the excision of the plasmid body and the *rosy* marker gene. The plasmid that was injected into the *Fab-7^attp50^* line contains an *attB* site for integration, *frt* sites for excision of the *rosy* gene, and *lox* sites for excision of the plasmid body. Testing elements were inserted just in front of the *attB* site. After the integration of the plasmid within *Fab-7^attp50^*, *ry+* transformants were selected. Then, *rosy* and plasmid cassettes were excised by FLP-recombinase to remove about 10.2 kb additional sequence between the tested element and *iab-7* in the *ry^+^* line.

In the *F2^attP^* platform [14], the 2106 bp within the *Fub* region (genome release R6.22: 3R:16,797,757 to 16,799,862 or complete sequence of BX-C in SEQ89E numbering: 183,576 to 185,681) was substituted with [*attP*]-[*lox*] sites. For the *F2^attP^* replacement, the recombination plasmid contains several genetic elements in the following order: [*attB*]-[*polylinker*]-[*lox*]-[*3P3-mCherry*]-[*mini-y*]. All elements were assembled within the *pBluescript SK* vector. *loxP* site is located after the *polylinker* and, in combination with the second site, which is located in the platform, is used for the excision of marker genes and plasmid body. DNA fragments used for the replacement experiments were generated by PCR amplification and verified by sequencing. The sequences of the used fragments are shown in Appendix A. The polyadenylation signal (*PAS*) corresponds to 700 bp *Xba*I-*Bam*HI fragment of main SV40 termination sequences from the pUAST base vector.

### 4.2. Cuticle Preparations

Three-day adult flies were collected in Eppendorf tubes and stored in 70% ethanol for at least 1 day. Then, ethanol was replaced with 10% KOH, and flies were heated at 70 °C for 1 h. After, the heating flies were washed with dH_2_O two times and heated again in dH_2_O for 45 min. Then, the digested flies were washed with 70% ethanol and stored in 70% ethanol. The abdomen cuticles were cut from the rest of the digested fly using a fine tweezer and a needle of an insulin syringe and put in a droplet of glycerol on a glass slide. Then, the abdomens were cut longitudinally on the dorsal side through all of the tergites with the syringe. To spread the cuticles flat on the slides, cuts may be carried out between the tergites. Then, the cuticles were flattened with a coverslip. Photographs in the bright or dark field were taken on the using Nikon DS-Ri2 digital camera (Nikon microscope products, Tokyo, Japan, 2015), processed with ImageJ 1.50c4 and Fiji bundle 2.0.0-rc-46.

## Figures and Tables

**Figure 1 ijms-24-11368-f001:**
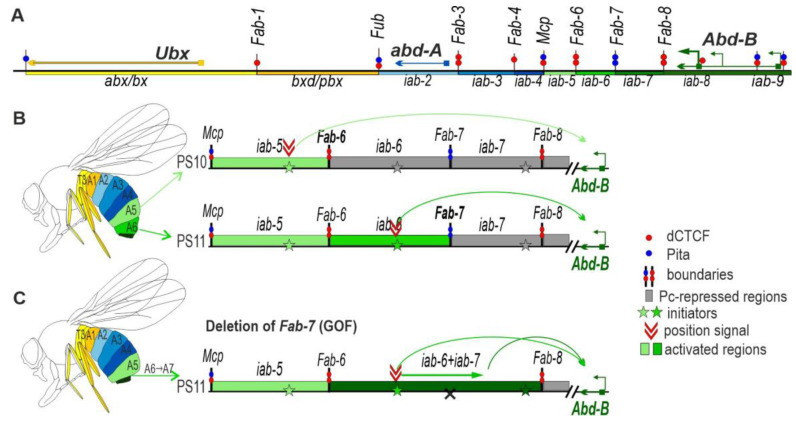
Boundaries organize regulatory region of the BX-C. (**A**) Map of the BX-C showing the location of the three homeotic genes and the parasegment-specific regulatory domains. There are nine *cis*-regulatory domains (shown as colored boxes) that are responsible for the regulation of the BX-C genes and the specification of parasegments 5 to 13, which correspond to T3-A8 segments (shown on the diagram of the adult fly using the same color code). The *abx/bx* (yellow) and *bxd/pbx* (orange) domains activate *Ubx*, *iab-2–iab-4* (shades of blue)—*abd-A* and *iab-5–9* (shades of green)—*Abd-B*. Lines with colored circles mark chromatin boundaries. The dCTCF and Pita binding sites at the boundaries are shown as red and blue circles, respectively. (**B**) Schematic presentation of *Abd-B* activation in A5(PS11) and A6(PS12) segments (parasegments). (**C**) Deletion of the *Fab-7* boundary results in premature activation of the *iab-7* domain in A6 (PS12).

**Figure 2 ijms-24-11368-f002:**
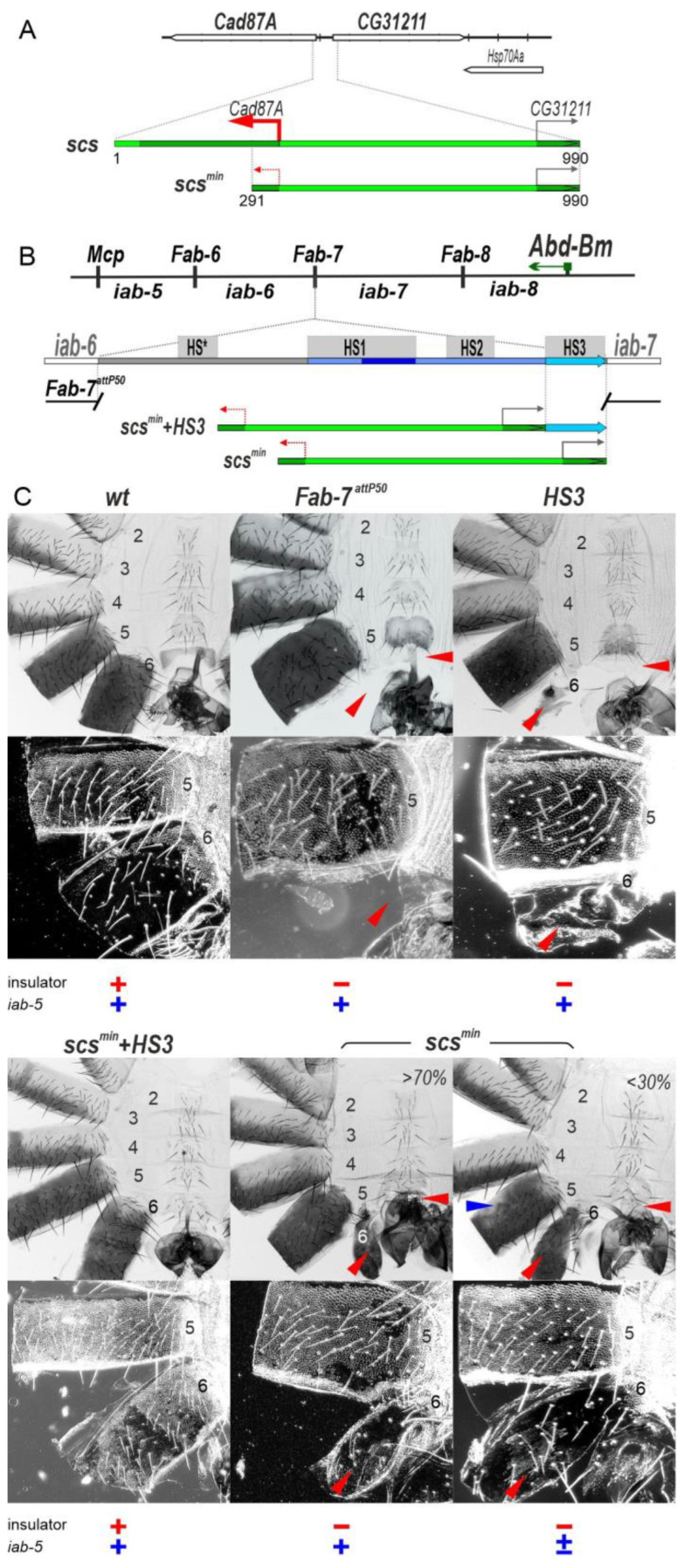
HS3 (*iab-7* PRE) is required for the boundary activity of the *scs^min^* insulator. (**A**) Schematic presentation of the *scs* insulator (990 bp) and *scs^min^* (700 bp) marked as green lines. TSS is the transcription start sites of *Cad87A* and *CG31211* are marked by red and grey arrows, respectively, indicating the orientation of the promoters. The truncated *Cad87A* promoter is marked by the dotted red arrow. Exons of *Cad87A* and *CG31211* are marked as dark green boxes. (**B**) Schematic representation of the *abd-A*–*Abd-B* regulatory regions and *Fab-7^attP50^* platform in which four hypersensitive sites, HS*, HS1, HS2, and HS3 (marked as grey boxes), are deleted. The endpoints of the deletion are indicated by breaks in the line. HS3 marked as azure arrow. Replacement fragments are shown below the map. (**C**) Morphology of the male abdominal segments (numbered) in *wild type* (*wt*), *Fab-7^attP50^*, *HS3* [46], *scs^min^+HS3*, and *scs^min^* transgenic lines. Bright-field (top) and dark-field (bottom) images of cuticles prepared from males. The filled red arrowheads show morphological features indicative of boundary inactivation (GOF phenotype). The blue arrowheads show signs of the *iab* enhancer inactivation (LOF phenotype). The insulator function is indicated by red “+” (normal) and “−” (moderate or strong inactivation). The function of the *iab-5* enhancers is shown by blue “+” (normal), “+/−” (weak inactivation), “−” (strong inactivation). In wild type, the A5 and A6 tergites (on the dorsal side of the fly) are fully pigmented, while the A2–4 tergites have only a stripe of pigmentation. The trichome hairs (shown in dark field) on the A6 tergite are restricted to the anterior and ventral margins, while A5 tergite is typically covered in uniform lawn trichomes (although there are small patches of cuticle that have only a few trichomes). The more anterior tergites are fully covered in trichome hairs, and the hairs are much more densely packed. The A2–A5 sternites on the ventral side have a quadrilateral shape with multiple bristles. The A6 sternite is different in that it lacks bristles and has a wide and curved shape. In males, the A7 segment is absent. Descriptions of other phenotypes are found in the text.

**Figure 3 ijms-24-11368-f003:**
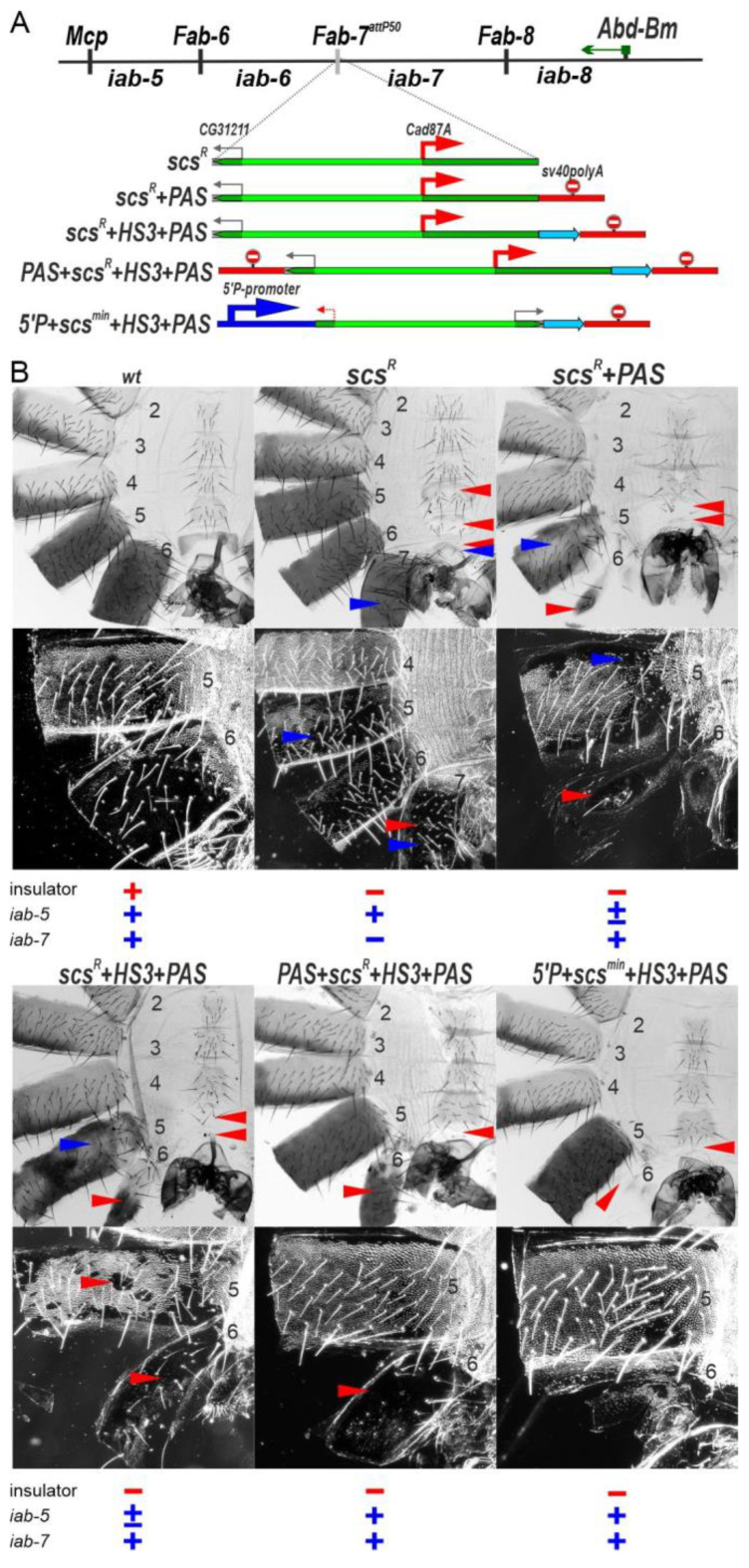
The *Cad87A* promoter in *scs* inserted in the reverse orientation (*scs^R^*) is responsible for inactivation of the *iab-7* domain. (**A**) Schemes of *Fab-7* boundary replacement with *scs* inserted in reverse orientation. The P-element promoter is shown as blue arrow. Polyadenylation signal from *SV40* (PAS) is shown as red line with stop signal. Other designations are as in Figure 2. (**B**) Morphology of the male abdominal segments (numbered) in *wt*, *scs^R^*, *scs^R^+PAS*, *scs^R^+HS3+PAS*, *PAS+scs^R^+HS3+PAS,* and *5′P+scs^min^+HS3+PAS*. The filled red arrowheads show morphological features indicative of boundary inactivation (GOF phenotype). The blue arrowheads show signs of the *iab* enhancer inactivation (LOF phenotype).

**Figure 4 ijms-24-11368-f004:**
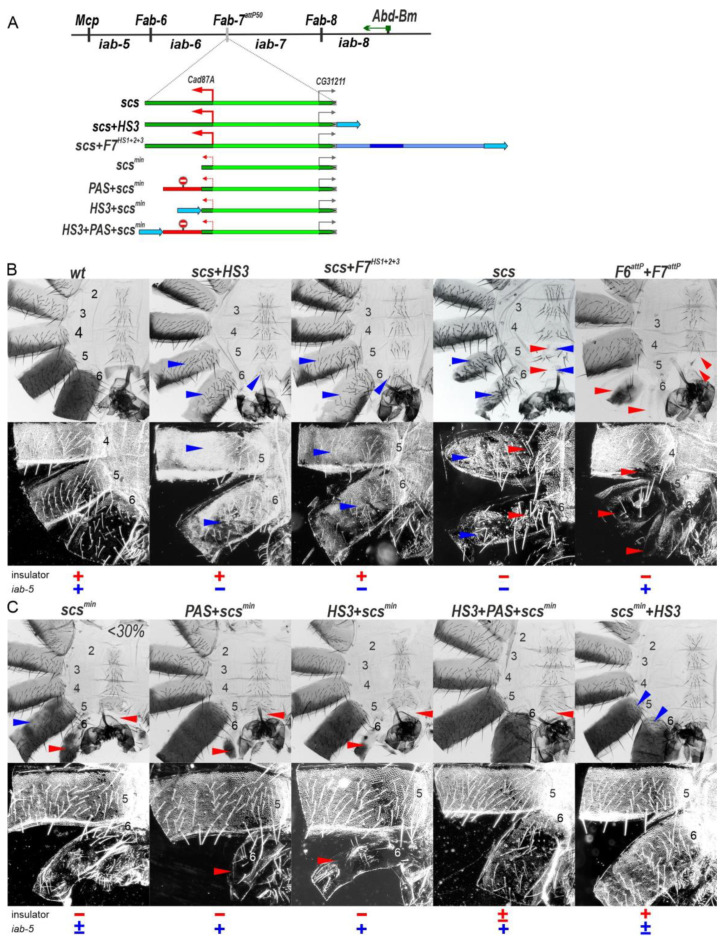
The *Cad87A* promoter in the *scs* inserted in the direct orientation (*scs*) is responsible for inactivation of the *iab-5* and *iab-6* domains. (**A**) *Fab-7* boundary replacement schemes in which *scs* and *scs^min^* were inserted in direct orientation in different combinations with the *Fab-7* fragments and PAS. Designations are as in Figure 2 and Figure 3. (**B**) Bright-field (top) and dark-field (bottom) images of cuticles prepared from males of *wt*, *scs, scs+HS3*, *scs+F7*, and *F6^attP^+F7^attP^*. (**C**) Comparing images of cuticles prepared from males of *scs^min^, PAS+scs^min^, HS3+scs^min^* and *HS3+PAS+scs^min^*. Bright-field (top) and dark-field (bottom) The filled red arrowheads show morphological features indicative of boundary inactivation (GOF phenotype). The blue arrowheads show signs of the *iab* enhancer inactivation (LOF phenotype).

**Figure 5 ijms-24-11368-f005:**
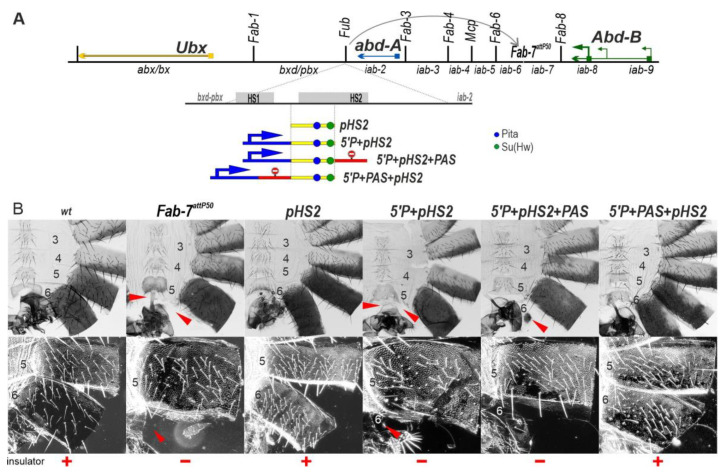
Transcription from the P-element promoter suppresses the activity of the *Fub* subfragment *pHS2* when it replaces the *Fab-7* boundary. (**A**) *Fab-7* boundary replacement schemes, in which *pHS2* was inserted in different combinations with the P-element promoter and PAS. Designations are as in Figure 2 and Figure 3. (**B**) Bright-field (top) and dark-field (bottom) images of cuticles prepared from males of *wt*, *Fab-7^attP50^*, *pHS2*, *5′P+pHS2+PAS*, and *5′P+PAS+pHS2.* The filled red arrowheads show morphological features indicative of boundary inactivation (GOF phenotype).

**Figure 6 ijms-24-11368-f006:**
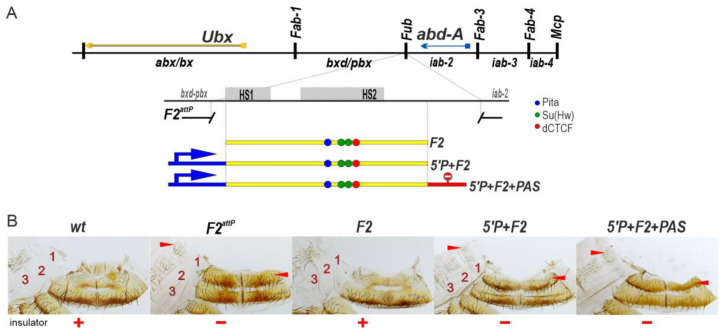
Transcription from the P-element promoter suppresses the functional activity of the *Fub* boundary and the *Fub* subelement *pHS2* in its endogenous location between the *bxd*/*pbx* (the *Ubx* regulatory region) and *iab-2* (the *abd-A* regulatory region) domains. (**A**) *Fub* boundary replacement schemes, in which *Fub* was inserted in different combinations with the P-element promoter and PAS. Designations are as in Figure 2 and Figure 3. (**B**) Morphology of the male abdominal segments (numbered) in *wt*, *F2^attP^*, *F2*, *P+F2*, and *5′P+F2+PAS*. The filled red arrowheads show morphological features indicative of boundary inactivation (GOF phenotype).

## Data Availability

Data are contained within the article or Appendix A.

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
