# Peer review of "Transcriptional Readthrough Interrupts Boundary Function in Drosophila"

_ijms, 2023, doi:10.3390/ijms241411368_

Round 1

Reviewer 1 Report

This manuscript describes an impact of readthrough transcription on boundary function of chromatin. The experiments are carefully executed, and the manuscript is beautifully documented. I have no serious concerns, but only have a few minor suggestions to improve clarity.

1. Fig. 1: The label C is missing in the figure. In the text (lines 88-90), it would be better to cite Figure 1B and 1C as follows: For example, the Fab-7 boundary in the Abd-B region of BX-C separates the iab-6 and iab-7 domains (Figure 1B). When Fab-7 is deleted iab-6 and iab-7 fuse into a single domain and the iab-6 initiation element inappropriately activates iab-7 in PS11 (A6) (Figure 1C).

2. Fig. 2 legend: In (A), it says “transcription start sites are marked by magenta arrows”, but no magenta arrows in the figure. Also “PAS is designated as a STOP signal” does not make sense, and probably can be omitted. In (B), it says “the endpoints of the deletion are indicated by breaks in the red line”, but no red line can be seen in the figure. In (C), it is optional, but it would be easier to read if the order is changed to “wild type, Fab-7attP50, HS3, scsmin+HS3, scsmin transgenic lines” in the order of appearance in the figure.

Author Response

“ Fig. 1: The label C is missing in the figure. In the text (lines 88-90), it would be better to cite Figure 1B and 1C as follows: For example, the Fab-7 boundary in the Abd-B region of BX-C separates the iab-6 and iab-7 domains (Figure 1B). When Fab-7 is deleted iab-6 and iab-7 fuse into a single domain and the iab-6 initiation element inappropriately activates iab-7 in PS11 (A6) (Figure 1C).”

The label C was inserted in the figure.

“Fig. 2 legend: In (A), it says “transcription start sites are marked by magenta arrows”, but no magenta arrows in the figure. Also “PAS is designated as a STOP signal” does not make sense, and probably can be omitted. In (B), it says “the endpoints of the deletion are indicated by breaks in the red line”, but no red line can be seen in the figure. In (C), it is optional, but it would be easier to read if the order is changed to “wild type, Fab-7attP50, HS3, scsmin+HS3, scsmin transgenic lines” in the order of appearance in the figure.”

We made all changes in Figure legend suggested by the reviewer. We thank the referee for drawing attention to mistakes in the figure legends. We carefully checked the figure legends and references for the figures in the text of the manuscript.

Reviewer 2 Report

The manuscript ijms-2459890, entitled “Transcriptional read through interrupts boundary function in Drosophila” authored by Kyrchanova et al., is aimed at the understanding of several aspects of chromatin boundary functions. The authors have selected Drosophila melanogaster as the organism of choice for their study. This selection presents some obvious advantages, especially because they centred their research in the well-known Bithorax complex. The subject of the research is interesting as many of the functions of insulator and boundary elements in chromatin are still unknown. In this way, the research might be of interest for an ample readership working in chromatin and in its 3D organisation. Nevertheless, this interest is limited by the fact that the authors have written their manuscript in a style easily understandable for researchers in Drosophila genetics, but hardly readable by other chromatin specialists.

The conclusions drawn from the experiments are solid, but they are based solely on the phenotypic consequences of the different constructs. Transcription of different genes is inferred from the changes in phenotype and the results are conditioned by this inductive reasoning.

To increase the potential audience of the present research, the manuscript must be entirely re-written trying to convey how the results are interesting in a wider chromatin structure scenario. In many sections, the specific Drosophila jargon must be explained.

Molecular validation of some of the results, especially those concerning the appearance of readthrough transcripts would be desirable. The authors have used molecular methods in some other cases (e.g., in reference 14) and acquiring these data might be an affordable question.

The Materials and Methods section is clearly insufficient. It is mentioned in section 4.2 that cuticle preparations were prepared essentially as described in [40]. The underlined adverb suggests that some changes in the protocol were done and, therefore, the method had to be detailed. Expanding the section 4.1 is even more necessary.

Some minor details may be considered. For instance, lettering of different panels is missing in some figures (C in fig. 1 and fig. 2); the caption to fig. 2B refers to the “endpoints of the deletion are indicated by breaks in the red line”, and no red line is drawn.

Author Response

“The conclusions drawn from the experiments are solid, but they are based solely on the phenotypic consequences of the different constructs. Transcription of different genes is inferred from the changes in phenotype and the results are conditioned by this inductive reasoning.”

The BX-C locus has a very complex tissue-specific regulation of expression. Therefore, an accurate study of tissue-specific transcription is an extremely difficult task. The aim of this work was to demonstrate the role of transcription in the inactivation of border activity. For this purpose, the SCS and P element promoters were used, the activity of which had previously been studied within the framework of the complex. To terminate transcription, the most effective PAS (from SV40) was used, which blocks all transcription that takes place in exons and introns.

“To increase the potential audience of the present research, the manuscript must be entirely re-written trying to convey how the results are interesting in a wider chromatin structure scenario. In many sections, the specific Drosophila jargon must be explained.”

Thanks to the reviewer for this comment. This article is written in the traditional style, which is accepted for the scientists working in the field of Drosophila genetics. A detailed description of the change in the phenotype provides information about the activity of the iab-5, iab-6 and iab-7 enhancers and boundary function between the iab-6 and iab-7 domains. We agree that understanding the significance of these genetic results is a big problem for scientists especially those who do not study Drosophila.  Unfortunately, there is no easy way to simplify the description of the phenotypes etc in a way that would still be meaningful.  To help the readers understand the nature of the phenotypic transformations and their implications in different replacement genotypes we have added a description of the cuticle morphology in each segment in wild type males to the legend of Fig. 2.  We also inserted in all figures short explanation of the results: presence/absence of the insulator activity and functional activity of the iab-5 and iab-7 enhancers.

 “Molecular validation of some of the results, especially those concerning the appearance of readthrough transcripts would be desirable. The authors have used molecular methods in some other cases (e.g., in reference 14) and acquiring these data might be an affordable question.”

The reviewer suggests using qChIP to study the binding of architectural/insulator proteins to boundaries, as was done in the mentioned reference 14. This would be a great addition for two reason.  First,  we would be able to find out/show that readthrough transcription adversely impacts the binding interactions of architectural proteins with boundaries. Second, it would be step in addressing problem mentioned in the previous point—making the work accessible to a larger audience.  Experiments of this type were tried and unfortunately they didn’t give an unambiguous result.  The problem is that readthrough transcription is expected to be limited to only the group of cells in which the regulatory domain is active.  For an Fab-7 replacement, that would be PS11-PS13/14. In the cells anterior to PS11, the promoter would be off and the architectural proteins won’t be displaced.  Even in PS11-PS13/14 cells it is possible that readthrough will be happing in only a subset of cells at end one time. This will depend on the frequency of transcriptional bursts. To address this issue, we are currently preparing new model systems for a detailed study of the mechanisms of action of transcription on the activity of boundaries/insulators.

The Materials and Methods section is clearly insufficient. It is mentioned in section 4.2 that cuticle preparations were prepared essentially as described in [40]. The underlined adverb suggests that some changes in the protocol were done and, therefore, the method had to be detailed. Expanding the section 4.1 is even more necessary.

We added the required information to Material and Methods section.

“Some minor details may be considered. For instance, lettering of different panels is missing in some figures (C in fig. 1 and fig. 2); the caption to fig. 2B refers to the “endpoints of the deletion are indicated by breaks in the red line”, and no red line is drawn.”

Fixed

Round 2

Reviewer 2 Report

The authors have addressed the questions I posed in my first report. Some of them, especially those dealing with the content of the Materials and Methods section, have been properly answered and the authors have expanded the text accordingly. The description of the cuticle morphology inserted in the caption to fig. 2, as well as some other additions in other figure legends make the text more understandable to non-specialised readers.

The “minor details” I mentioned have been correctly dealt with.

Some of my recommendations have not resulted in modifications of the manuscript, but I understand the reasons given by the authors not to do so.